# Antimicrobial Resistance of *Actinobacillus pleuropneumoniae*, *Streptococcus suis*, and *Pasteurella multocida* Isolated from Romanian Swine Farms

**DOI:** 10.3390/microorganisms11102410

**Published:** 2023-09-27

**Authors:** Madalina Iulia Siteavu, Roxana Ionela Drugea, Elena Pitoiu, Emilia Ciobotaru-Pirvu

**Affiliations:** 1Faculty of Veterinary Medicine, University of Agronomic Sciences and Veterinary Medicine of Bucharest, 050097 Bucharest, Romania; 2Synevovet Laboratory, Ilfov County, 077040 Chiajna, Romania

**Keywords:** *Actinobacillus pleuropneumoniae*, *Streptococcus suis*, *Pasteurella multocida*, antimicrobial resistance, swine

## Abstract

Antimicrobial resistance is an important health issue in human and veterinary medicine. The aim of this study was to monitor the antimicrobial resistance of three of the most important bacteria involved in porcine respiratory disease. A total of 465 isolates were tested during the 2017–2022 period for antimicrobial susceptibility for *Actinobacillus pleuropneumoniae* (n = 137), *Streptococcus suis* (n = 207), and *Pasteurella multocida* (n = 121) by disk diffusion method. The results were interpreted by CLSI breakpoints, where available. High rates of susceptibility (from 90 to >99%) were observed for cefquinome, ceftiofur, amoxicillin + clavulanic acid, amoxicillin, penicillin, ampicillin, florfenicol, enrofloxacin, marbofloxacin, and trimethoprim–sulfamethoxazole. *A. pleuropneumoniae* isolates showed high resistance to streptomycin (77%), gentamycin (45%), tilmicosin (39%) erythromycin (33%), oxytetracycline (19%), and tetracycline (18%). For *S. suis*, the highest rates of resistance were observed for streptomycin (98%), tetracycline (75%), oxytetracycline (72%), doxycycline (52%), and erythromycin (51%). *P. multocida* presented a high rate of resistance for streptomycin (63%), tilmicosin (29%), oxytetracycline (13%), and tetracycline (14%). Bacteria isolates maintained high susceptibility against antimicrobial agents usually used against the mainly respiratory tract pathogens of swine. Resistance for streptomycin, tetracycline, oxytetracycline, and tilmicosin was high for all the tested pathogens

## 1. Introduction

The porcine respiratory disease complex (PRDC) is the result of the interaction of many factors, being a typical polymicrobial production disease and one of the most significant health and production problems, considered to be a leading cause of economic losses in the swine industry [1,2,3,4]. Several factors contribute to the development of respiratory disease along with viral and bacterial infections, such as genetics, environmental conditions, production system, or management [1,2]. A variety of bacterial pathogens is associated with PRDC, primary pathogens, such as *Mycoplasma hyopneumoniae*, *Actinobacillus pleuropneumoniae*, and *Bordetella bronchiseptica,* and secondary pathogens, including *Pasteurella multocida*, *Streptococcus suis*, *Glaeserella parasuis*, *Actinobacillus suis*, and *Salmonella* spp. [1,3,5,6]. 

Respiratory disease management is a major source of antibiotic use in the swine industry. It has been estimated that 60% of swine nurseries and 70% of the swine grower–finisher sites use injectable antibiotics to treat respiratory disease in the USA, while in Canada, approximately 44% to 67% of the grower–finisher pigs received antimicrobials in feed for preventive purposes [7,8,9].

*Actinobacillus pleuropneumoniae* (*A. pleuropneumoniae*) is a Gram-negative, facultative anaerobic bacteria and the causative agent of porcine pleuropneumonia, one of the most important life-threatening respiratory diseases in pigs, the most virulent strains determining fatal fibrinohemorrhagic and necrotizing pleuropneumonia in swine of all ages and chronic lung lesions in survivors [6,10,11,12]. Antimicrobials are the most effective control measure of *A. pleuropneumoniae*, leading to increasing levels of acquired resistance [10]. 

*Streptococcus suis* (*S. suis*) causes pneumonia and a diverse range of septic diseases, such as meningitis, arthritis, valvular endocarditis, polyserositis, abortion, and sudden death, mostly in the 5–10-week-old pigs [6,11,13]. When antimicrobials are being used as a prophylactic or metaphylactic measure, the incidence of disease is generally lower than 5%; otherwise, the mortality may rise to 20%, but the antimicrobials that are efficient against S. suis are those that the industry is trying to reduce the usage of, because of their importance in both human and veterinary medicine [13]. 

*Pasteurella multocida* (*P. multocida*) is among the most commonly found and costly bacteria involved in PRDC, causing progressive atrophic rhinitis, with the destruction of the turbinate bones, and pneumonia [6,11,13]. It is a commensal bacteria of the upper respiratory tract that can also cause pneumonia in growing and finishing pigs. *P. multocida* is normally considered a secondary agent, but it has also been described as a primary agent of hemorrhagic septicemia in pigs [14]. Antimicrobials are the most used veterinary products for the management of *P. multocida* in animals including first-generation antibiotics [15].

Increasing antimicrobial resistance for many microorganisms has been reported in the past decades, so the interest in monitoring the trends of antimicrobial susceptibility of veterinary pathogens has extended [16]. The aim of this study was to evaluate the antimicrobial susceptibility evolution of *A. pleuropneumoniae*, *S. suis*, and *P. multocida* isolated from swine in Romania from 2017 to 2022.

## 2. Materials and Methods

### 2.1. Sample Collection

This study was conducted on 465 bacterial isolates, 137 for *A. pleuropneumoniae*, 207 for *S. suis*, and 121 for *P. multocida*, as seen in Table 1, recovered from samples collected from lung tissue of swine with respiratory disease. Usually, the samples are collected with swabs by the veterinarian from the farm, and the administration of antibiotics and types of antibiotics given are not constantly reported. 

The bacterial isolates included in this study were obtained in a six-year period, between 2017 and 2022, in the Synevovet laboratory from Romania, and the farms are located all over the country, mainly in the south and west areas (Figure 1).

### 2.2. Isolation and Identification

The samples were cultured on chocolate agar and blood agar mediums and then incubated at 35–37 °C for 20–24 h in anaerobic conditions (CO_2_ 5% thermostat, MMM Group, Munich, Germany). The colonies were selected based on their morphological characteristics, and then the pathogens were identified using MALDI-TOF technology (Matrix-assisted laser desorption/ionization time-of-flight mass spectrometry, Bruker Daltonics, Bremen, Germany).

### 2.3. Antimicrobial Susceptibility Testing

Antimicrobial susceptibility was tested using the disk diffusion method on Mueller–Hinton agar supplemented with 5% sheep blood (for *S. suis* and *P. multocida*) and chocolate agar (for *A. pleuropneumoniae*) [17,18]. The inhibition zone sizes around the antibiotic disks were read with an ADAGIO automated system (Bio-Rad, Marnes-la-Coquette, France, 3.1). The interpretation of the resistance profiles was performed according to the CLSI (Clinical and Laboratory Standards Institute) standards, if available, and the antibiotic manufacturer. The following antimicrobials and concentrations were used and then classified as susceptible, intermediate, and resistant, as seen in Table 2. 

## 3. Results

The mean antibiotic susceptibility of *A. pleuropneumoniae*, *S. suis*, and *P. multocida* over the 6-year tested period is shown in Table 3. High rates of susceptibility (from 90% to >99%) were observed for *A. pleuropneumoniae* for cefquinome (100%), ceftiofur (99%), amoxicillin + clavulanic acid (99%), amoxicillin (95%), penicillin (95%), ampicillin (94%), florfenicol (93%), enrofloxacin (92%), marbofloxacin (92%), and trimethoprim–sulfamethoxazole (91%). For *Streptococcus suis*, the highest rates of susceptibility were observed for ceftiofur (99%), amoxicillin + clavulanic acid (98%), florfenicol (97%), amoxicillin (95%), ampicillin (95%), cefquinome (93%), penicillin (92%), and marbofloxacin (91%). *P. multocida* showed high susceptibility for many of the tested antibiotics such as marbofloxacin (99%), amoxicillin + clavulanic acid (98%), ceftiofur (98%), amoxicillin (97%), penicillin (97%), florfenicol (97%), ampicillin (96%), enrofloxacin (96%), gentamycin (95%), and trimethoprim–sulfamethoxazole (93%).

*A. pleuropneumoniae* isolates showed the lowest rates of resistance to the tested pathogens. High resistance to streptomycin (77%), gentamycin (45%), tilmicosin (39%), erythromycin (33%), oxytetracycline (19%), and tetracycline (18%) was observed. *S. suis* showed the highest rates of resistance to the studied pathogens; the antibiotics with the highest rates being streptomycin (98%), tetracycline (75%), oxytetracycline (72%), doxycycline (52%), erythromycin (51%). *P. multocida* presented a high rate of resistance for streptomycin (63%), tilmicosin (29%), oxytetracycline (13%), and tetracycline (14%).

The resistance rates of *A. pleuropneumoniae* are reported in Table 4, and the trends of resistance over the study period can be observed in Figure 2. *A. pleuropneumoniae* resistance for penicillin, tetracycline, trimethoprim–sulfamethoxazole, enrofloxacin, marbofloxacin, tilmicosin, reached the highest level in 2020, followed by a decrease in the following years. Streptomycin and gentamycin resistance rates were at the highest level in 2021, with lower prevalence next year, while oxytetracycline and doxycycline showed increasing trends over the tested period.

Antibiotic resistance of *S. suis* decreased over time for the antibiotics with the highest resistance rates, for tetracycline from 95% to 63%, for oxytetracycline from 86% to 63%, and for doxycycline from 76% to 44%. For erythromycin, the level of resistance remained high over the years, although the resistance rates were lower in 2022 compared with 2017. The resistance rates of *S. suis* over the tested period are presented in Table 5, and the trends of resistance are shown in Figure 3.

For *P. multocida,* antibiotic resistance rates are presented in Table 6, and the trends of resistance over the study period are shown in Figure 4. For streptomycin and tilmicosin, the resistance was high and inconstant, varying from 100% to 45% and 50% to 21%, respectively. The resistance to tetracycline and oxytetracycline shows a decrease over the years, reaching, respectively, 4% and 3% in 2022 from 25% and 29% in 2019.

## 4. Discussion

Following the extent of use in human and veterinary medicine, antibiotic-resistant microorganisms are contaminating the soil and water, and their pattern of resistance is correlated with the degree of antibiotic use [19]. Because of the urgent need for antimicrobial treatment in acute infections, the treatment is based very often only on the susceptibility pattern of bacterial pathogens to antibiotics [11]. 

Numerous retrospective studies have been published for antibiotic susceptibility testing for *A. pleuropneumoniae*, *S. suis*, and *P. multocida* across the world. 

### 4.1. Actinobacillus pleuropneumoniae

In our study, 90 to more than 99% susceptibility was observed for *A. pleuropneumoniae* for cefquinome (100%), ceftiofur (99%), amoxicillin + clavulanic acid (99%), amoxicillin (95%), penicillin (95%), ampicillin (94%), florfenicol (93%), enrofloxacin (92%), marbofloxacin (92%), and trimethoprim–sulfamethoxazole (91%). The results are in agreement with many of the data previously published. In the VetPath study (2015) conducted in 11 European countries, a high level of susceptibility was also reported for amoxicillin + clavulanic acid (100%), ceftiofur (100%), and trimethoprim–sulfamethoxazole (96.8%). [16] The susceptibility to tiamulin (100%) and tilmicosin (99.4%) was much higher in this study than our results of 60% and 58%, respectively. High susceptibility to most antimicrobial agents for these pathogens was also reported in a study of swine respiratory disease from the United States and Canada. *A. pleuropneumoniae* isolates were 100% susceptible to ceftiofur and florfenicol and 90 to >99% to enrofloxacin and tulathromycin [5,20]. In a study from Spain, from 2017 to 2019, *A. pleuropneumoniae* were highly susceptible (≥90%) to tildipirosin, tulathromycin, tilmicosin, tiamulin, florfenicol, sulfamethoxazole/trimethoprim, and ceftiofur. The antimicrobial susceptibility was intermediate (72%) for amoxicillin and enrofloxacin and low (35.7%) for doxycycline [21], much lower for these three antibiotics compared with our study. 

Our results showed high resistance of *A. pleuropneumoniae* isolates to streptomycin (77%), gentamycin (45%), tilmicosin (39%), erythromycin (33%), oxytetracycline (19%), and tetracycline (18%). In a study from Australia, the resistance of *A. pleuropneumoniae* isolates to erythromycin (89%) and tetracycline (75%) was higher than observed in our study. Resistance to ampicillin (8.5%), penicillin (8.5%), and tilmicosin (25%) was also identified [1]. The resistance to tetracycline varied very much between different studies. Low susceptibility rates of *A. pleuropneumoniae* to tetracycline, 0 to 6%, were reported in another study [5]. In a Danish study, *A. pleuropneumoniae* displayed high values of resistance to erythromycin, and a few isolates were resistant to tetracycline. Almost all isolates were susceptible to the other antibiotics [22].

In a study of *A. pleuropneumoniae* conducted in Italy from 1994 to 2009, resistance to amoxicillin, amoxicillin/clavulanic acid, ampicillin, cefquinome, cotrimoxazole, penicillin G, and tilmicosin increased over time, while resistance to gentamycin and marbofloxacin decreased. The levels of resistance reported in the last year included in this study (2009) have been much higher than our results for many antibiotics such as amoxicillin (82.6%), ampicillin (69.2%), penicillin G (72.7%), cefquinome (23.8%), doxycycline (25%), and tetracycline (58.8%) [10]. In a study from China for *A. pleuropneumoniae*, there were six antibiotics with associated resistance rates of 50% or higher: streptomycin (53.57%); amikacin (53.57%); kanamycin (53.57%); tilmicosin (85.71%); lincomycin (96.43%); and compound trimoxazole (57.14%). Cephalosporins had the highest sensitivity, with resistance rates of 0% for cephalothin, 7.14% for ceftiofur, and 3.57% for cefotaxime [23]. This report shows a higher resistance to tilmicosin compared to our study. In another study from Korea, the tested *A. pleuropneumoniae* isolates were perfectly susceptible to amoxicillin/clavulanic acid, cephalothin, and ceftiofur, while the isolates were resistant to lincomycin and erythromycin. Florfenicol’s susceptibility decreased from 2006 (94%) to 2010 (50%), and a minor decrease was observed in ampicillin and amoxicillin. Ampicillin and amoxicillin were active against over 88% of *A. pleuropneumoniae* isolates in 2007, while in 2010, only 65% showed susceptibility [12].

In a study from the Czech Republic, 242 *A. pleuropneumoniae* isolates showed low levels of antimicrobial drug resistance, except for tetracycline (23.9%). Resistance was low to florfenicol (0.8%), tiamulin (1.7%), tilmicosin (1.2%), ampicillin (3.3%), and amoxicillin/clavulanic (0.8%). Resistance to ceftiofur was not detected in any isolate [24]. The results are similar to ours, except also for tiamulin and tilmicosin, which show a higher level of resistance in Romania. In a previous study from Romania for antibiotic resistance to *A. pleuropneumoniae*, the results showed an increasing trend of resistance to cefaclor, amoxicillin/clavulanic acid, oxytetracycline, ampicillin, and enrofloxacin and high susceptibility for florfenicol and erythromycin [25].

*A. pleuropneumoniae* trends of resistance in our study showed that for six antibiotics (penicillin, tetracycline, trimethoprim–sulfamethoxazole, enrofloxacin, marbofloxacin, and tilmicosin), the highest levels were reached in 2020, followed by a decrease in the following years. Streptomycin and gentamycin resistance rates were at the highest level in 2021, with lower prevalence next year, while oxytetracycline and doxycycline showed increasing trends over the tested period. In a report from Japan, no significant increase in resistance to the antimicrobials tested in this study was observed during the last decade when compared with published data on the prevalence of resistant strains of *A. pleuropneumoniae* collected from 1989 to 2005 [26].

### 4.2. Streptococcus suis

In this study, for *S. suis*, the highest rates of susceptibility were observed for ceftiofur (99%), amoxicillin + clavulanic acid (98%), florfenicol (97%), amoxicillin (95%), ampicillin (95%), cefquinome (93%), penicillin (92%), and marbofloxacin (91%). *S. suis* has the highest rates of resistance to streptomycin (98%), tetracycline (75%), oxytetracycline (72%), doxycycline (52%), and erythromycin (51%). In the VetPath study, tetracycline resistance was 81.8% in *S. suis*, similar to our study, while the other tested antibiotics showed high susceptibility [16]. In a Danish study, *S. suis* showed a high level of resistance to erythromycin (47.1%) and tetracycline (77.3%) and a high susceptibility to penicillin, the results being similar to our study [22]. 

Antibiotic resistance of our tested isolates of *S. suis* decreased over time for tetracycline from 95% to 63%, for oxytetracycline from 86% to 63%, and for doxycycline from 76% to 44%. For erythromycin, the level of resistance remained high over the years, but still with lower rates in 2022 compared with 2017. Low susceptibility rates were observed for tetracycline, 0 to 1.3% for *S. suis* in a study from the United States and Canada [5], with a significant difference from our study. Another recent study conducted in 10 European countries showed that the resistance of *S. suis* to ceftiofur, enrofloxacin, and florfenicol was absent or lower than 5%, while resistance to tetracycline was 82.4% in *S. suis* [27].

In the VetPath study, a tendency to increase the resistance was observed for ceftiofur in isolates of *S. suis* [16]. In the Netherlands, it was reported that the resistance of *S. suis* to ampicillin, ceftiofur, clindamycin, enrofloxacin, florfenicol, penicillin, trimethoprim/sulfamethoxazole, and tetracycline was 0.3%, 0.5%, 48.1%, 0.6%, 0.1%, 0.5%, 3.0%, and 78.4%, respectively. In the same report, an increase in susceptibility to several antibiotics, similar to our study, was observed when comparing antimicrobial susceptibility for successive quarters. It was demonstrated that a significant decrease in susceptibility for clindamycin was observed in the fourth quarter of 2013 (compared with the third quarter of the same year), followed by a significant increase in the second quarter of 2014 (compared with the fourth quarter of 2013). The susceptibility of trimethoprim/sulfamethoxazole decreased significantly in the fourth quarter of 2013, compared with the third quarter of the same year, followed by a significant increase in the first quarter of 2014. For tetracycline, a significant decrease (*p* < 0.05) in susceptibility of *S. suis* was observed in the fourth quarter of 2013, compared with the third quarter of the same year and the second quarter of 2014, compared with the second quarter of 2013. Subsequently, a significant increase in tetracycline susceptibility of *S. suis* was seen in the fourth quarter of 2014. The reason for the decrease, followed by the increase in susceptibility for clindamycin, trimethoprim/sulfamethoxazole, as well as tetracycline, remains unknown [28]. 

In the United Kingdom, 405 *S. suis* isolates tested for antimicrobial susceptibility, and high rates of resistance were reported to tetracycline (91%), followed by erythromycin (46%), and trimethoprim/sulfamethoxazole (12%), the results being in similar to ours. The data showed a general trend of higher resistance between 2009 and 2011 and 2013 and 2014, in agreement with the previous European studies. A decrease in fluoroquinolone susceptibility in *S. suis* has been described. Subtle increases in resistance were found for ceftiofur and cefquinome between the first and second time periods [29]. In a study from China, *S. suis* was mostly resistant to apramycin (100%) and had the highest sensitivity to cephalothin (2.27%). Nine antibiotics were associated with resistance rates of 50% or higher: streptomycin (59.09%); gentamicin (90.91%); amikacin (97.73%); kanamycin (93.18%); apramycin (100%); erythromycin (68.18%); tilmicosin (97.73%); tetracycline (84.09%); and lincomycin (97.73%). *S. suis* had the highest sensitivity to cephalosporin antibiotics [23]. Tilmicosin, tetracycline, and erythromycin showed a higher level of resistance compared to our results.

### 4.3. Pasteurella multocia

*P. multocida* showed high susceptibility for many of the tested antibiotics such as marbofloxacin (99%), amoxicillin + clavulanic acid (98%), ceftiofur (98%), amoxicillin (97%), penicillin (97%), florfenicol (97%), ampicillin (96%), enrofloxacin (96%), gentamycin (95%), and trimethoprim–sulfamethoxazole (93%). High rates of resistance were found for streptomycin (63%), tilmicosin (29%), oxytetracycline (13%), and tetracycline (14%).

For *P. multocida*, the VetPath study revealed high susceptibility (from 90% to 100%) to amoxicillin + clavulanic acid, ceftiofur, enrofloxacin, tulathromycin, florfenicol, and tilmicosin. The only antibiotic with a high resistance rate was tetracycline (20.4%) [16]. Tilmicosin is the only antibiotic with lower resistance compared to our study. In a study of swine respiratory disease from the United States and Canada, *P. multocida* isolates were 100% susceptible to ceftiofur, enrofloxacin, and florfenicol and 90 to >99% to ampicillin, penicillin, tilmicosin, and tulathromycin. Low susceptibility rates were observed for tetracycline, from 22.3 to 35.3% [5]. Tilmicosin also showed a much higher susceptibility than in our study.

In a study from China, the results reveal that *P. multocida* was most resistant to lincomycin (100%), and it was sensitive to cephalothin, cefotaxime, apramycin, and doxycycline (100%). In our study, apramycin susceptibility was 50%. There were five antibiotics associated with a resistance rate of 50% or above: streptomycin (55.56%); gentamycin (50%); tilmicosin (77.78%); lincomycin (100%); and compound trimethoprim (61.11%). The doxycycline sensitivity rate was 100% [23]. These results show higher levels of resistance compared to our study for gentamycin, tilmicosin, and trimethoprim–sulfamethoxazole.

The antibiotic resistance trends of *P. multocida* for streptomycin and tilmicosin varied from 100% to 45% and from 50% to 21%, respectively. The resistance to tetracycline and oxytetracycline shows a decrease over the years, reaching 4% and 3% in 2022, from 25% and 29% in 2019, respectively. 

In a study conducted on 454 isolates of *P. multocida* collected from Korea between 2010 and 2016, the most frequently observed resistance was to sulphadimethoxine (76%), followed by oxytetracycline (66.5%), chlortetracycline (36.8%), and florfenicol (18.5%). No consistent increase or decrease in resistance was observed for most antimicrobials, except fluoroquinolones, whose resistance tended to increase over the study period [30]. In a study from Australia, the *P. multocida* isolates exhibited resistance to cotrimoxazole (2%), florfenicol (2%), ampicillin (4%), penicillin (4%), erythromycin (14%), and tetracycline (28%) [1]. In a study from Spain, from 2017 to 2019, *P. multocida* showed high susceptibility (≥90%) to tildipirosin, tulathromycin, tilmicosin, florfenicol, enrofloxacin, amoxicillin, and ceftiofur, intermediate (74.7%) resistance to sulfamethoxazole–trimethoprim and tiamulin (60.8%), and low (51.5%) for doxycycline [21]. In another study from Spain conducted on 48 *P. multocida* isolates, all isolates were susceptible to ceftiofur, florfenicol, tildipirosin, and tulathromycin, while most of them (>95%) were susceptible to amoxicillin, the two quinolones tested (enrofloxacin and marbofloxacin), and tilmicosin. Doxycycline and oxytetracycline showed resistance in 52.1% and 68.7% of the cases, respectively, which is much higher than our results. A total of 25% of isolates showed resistance to tiamulin and 31.2% or 43.7% to sulphamethoxazole/trimethoprim [14].

Since 2017, Romania has been the most African swine fever-affected country in the European Union, with 90% of the recorded outbreaks in domestic pigs [31]. The ongoing evolution of African swine fever led to many changes in the Romanian swine industry. There were more than 6000 outbreaks, and many farms were affected and went through stamping out, leading to a renewal of pig populations. Many farms took the opportunity to improve the management conditions and even repopulated with specific pathogen-free animals. This could explain the variation in results from recent years. The effect of stamping out measures on the bacteria prevalence in the farms and the evolution of antimicrobial resistance could be further studied.

Examining the results of the current study, it is important to understand that the data were collected from isolates submitted for bacteria identification, and a much larger study would be necessary for obtaining an insight into the national picture, with more information regarding farms’ antibiotic usage, clinical signs and lesions, vaccination programs, and strict sampling criteria.

Many European countries have developed surveillance systems for AMR in diseased animals, but antimicrobial resistance in animal bacterial pathogens is still considered a gap in the European One Health strategy on AMR surveillance [32]. In a review of the national monitoring systems for ABR in animal bacterial pathogens in Europe, 12 countries reported having a national monitoring system: the Czech Republic; Denmark; Estonia; Finland; France; Ireland; Germany; the Netherlands; Norway; Spain; Sweden; and the United Kingdom [33]. In Romania, there is no national surveillance program for antimicrobial susceptibility testing, even though many articles report an increase in bacteria resistance [25,34,35,36]. AMR is a priority public health issue. The country has one of the highest levels of antibiotic consumption in the human healthcare sector in Europe and faces some of the highest levels of antimicrobial resistance worldwide, as reported by the World Health Organization [37]. Even though some legislative measures were taken, such as establishing the National Committee for Prevention and Limitation of Healthcare-Associated Infections and elaborating the National Guidelines Regarding Prudent Use of Antimicrobials in Veterinary Medicine, monitoring of AMR in Romania should be coordinated at a national level and follow the European trend and recommendation. Also, it is necessary to harmonize the antibiotic susceptibility testing interpretation criteria within the European Union, and we should use EUCAST ECOFFs (European Committee on Antimicrobial Susceptibility Testing epidemiological cut-off values) in the future.

## 5. Conclusions

Bacteria isolates coming from Romanian swine farms, collected from 2017 to 2022, maintained high susceptibility against antimicrobial agents usually used against the mainly respiratory tract pathogens of swine, such as cefquinome, ceftiofur, florfenicol, amoxicillin + clavulanic acid, amoxicillin, penicillin, ampicillin, and fluoroquinolones. The resistance to streptomycin, tetracycline, oxytetracycline, and tilmicosin was high for all the tested pathogens. The responsible use of antimicrobials, according to laboratory diagnosis results, after antimicrobial susceptibility testing is advisable.

## Figures and Tables

**Figure 1 microorganisms-11-02410-f001:**
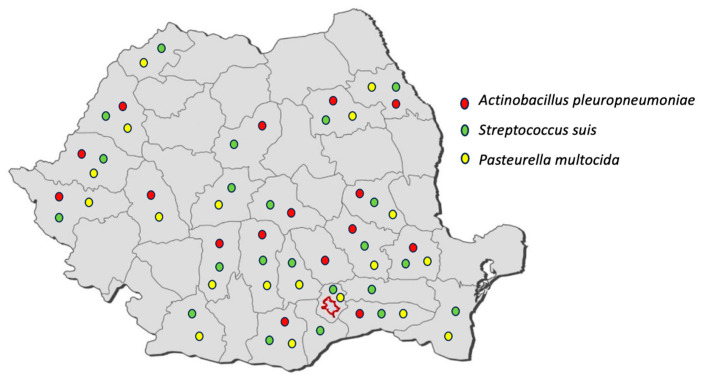
Map of Romania showing the geographic location of the farms from which the pathogens included in this study were identified.

**Figure 2 microorganisms-11-02410-f002:**
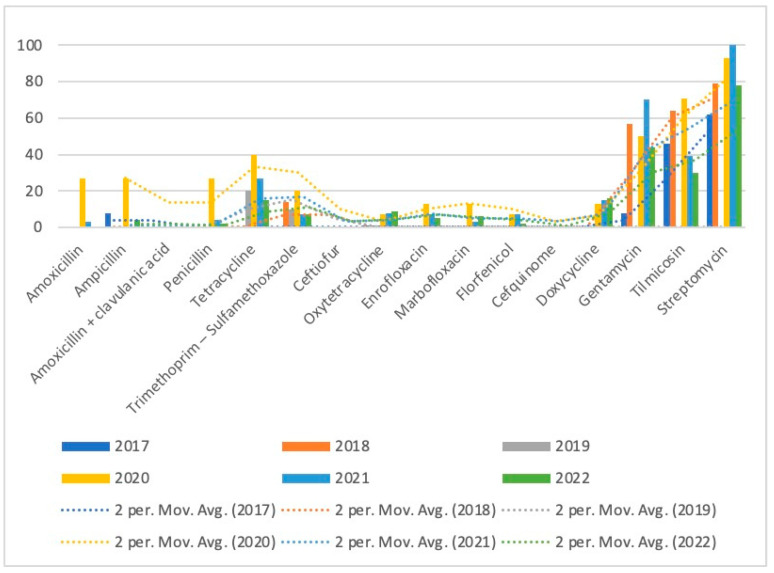
Trends of antimicrobial resistance of *A. pleuropneumoniae* isolated over 2017–2022.

**Figure 3 microorganisms-11-02410-f003:**
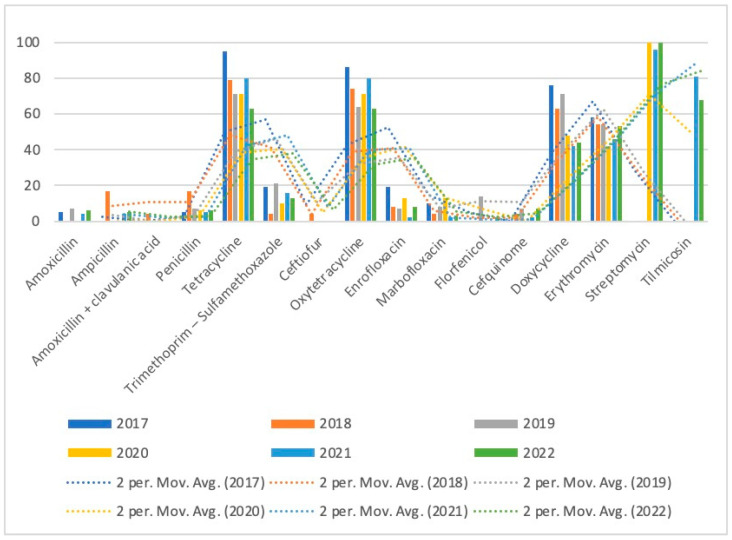
Trends of antimicrobial resistance of *S. suis* isolated over 2017–2022.

**Figure 4 microorganisms-11-02410-f004:**
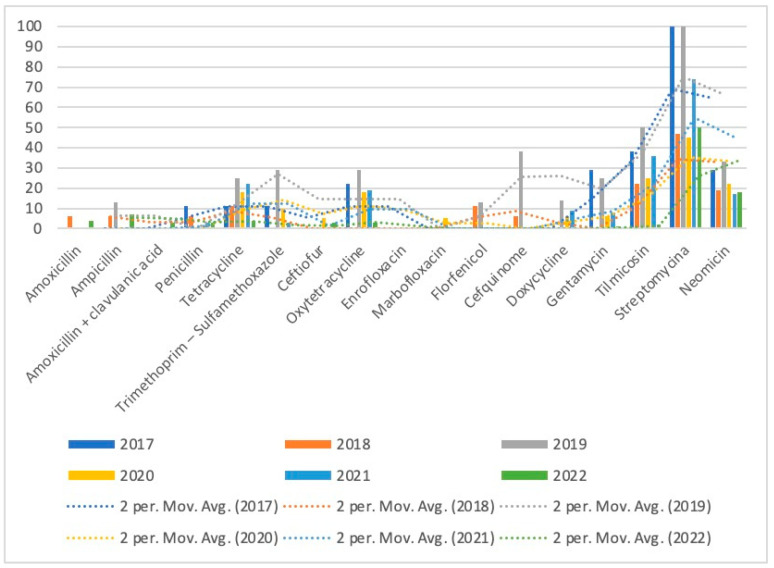
Trends of antimicrobial resistance of *P. multocida* isolated over 2017–2022.

**Table 1 microorganisms-11-02410-t001:** Number of isolates included in the study per year.

Title 1	2017 No. Isolates (%)	2018 No. Isolates (%)	2019 No. Isolates (%)	2020 No. Isolates (%)	2021 No. Isolates (%)	2022 No. Isolates (%)	Total
*APP **	13 (10)	14 (11)	10 (7)	15 (12)	30 (23)	55 (43)	*137*
*Streptococcus suis*	21 (10)	24 (12)	14 (14)	31 (15)	55 (27)	62 (30)	*207*
*Pasteurella multocida*	9 (7)	18 (15)	8 (14)	20 (17)	36 (30)	30 (25)	*121*

* *APP*—*Actinobacillus pleuropneumoniae*.

**Table 2 microorganisms-11-02410-t002:** Antimicrobials used for susceptibility testing, their concentration, and the inhibition diameters.

The Antimicrobial	Tablet Concentration (μg)	Susceptible (mm)			Intermediate (mm)			Resistant (mm)		
		*APP*	*S. suis*	*P. multocida*	*APP*	*S. suis*	*P. multocida*	*APP*	*S. suis*	*P. multocida*
**β-lactamase**										
Penicillin (P)	10	≥17	≥24	≥17	14–16	–	14–16	≤13	–	≤13
Ampicillin (AM)	10	≥17	≥24	≥17	14–16	–	14–16	≤13	–	≤13
Amoxicillin (AMX)	25	≥17	≥24	≥17	14–16	–	14–16	≤13	–	≤13
Amoxicillin + clavulanic acid (AMC)	20/10	≥18	≥18	≥18	14–17	14–17	14–17	≤13	≤13	≤13
Ceftiofur (CEF)	30	≥21	≥21	≥21	18–22	18–20	18–20	≤17	≤17	≤17
Cefquinome (CEQ)	30	≥33	≥22	≥22	20–21	20–21	20–21	≤19	≤19	≤19
**Quinolones**										
Enrofloxacin (ENR)	5	≥23	≥23	≥23	19–22	19–22	19–22	≤18	≤18	≤18
Marbofloxacin (MAR)	5	≥20	≥20	≥20	15–19	15–19	15–19	≤14	≤14	≤14
**Macrolides**										
Erythromycin (E)	15	–	≥21	–	–	16–20	–	–	≤15	–
Tilmicosin (TIL)	15	≥11	–	≥11	–	–	–	≤10	–	≤10
**Tetracyclines**										
Oxytetracycline (OT)	30	≥15	≥23	≥15	12–14	19–22	12–14	≤11	≤18	≤11
Tetracycline (TE)	30	≥15	≥23	≥15	12–14	19–22	12–14	≤11	≤18	≤11
Doxycycline (DOXY)	30	≥15	≥23	≥15	12–14	19–22	12–14	≤11	≤18	≤11
**Amphenicols**										
Florfenicol (FFC)	30	≥22	≥22	≥22	19–21	19–21	19–21	≤18	≤18	≤18
**Aminoglycosides**										
Streptomycin (S)	10	≥15	–	≥15	14	–	14	≤13	–	≤13
Gentamycin (GM)	10	≥16	–	≥16	13–15	–	13–15	≤12	–	≤12
Neomicin (N)	30	–	–	≥17	–	–	16	–	–	≤15
Apramycin (AP)	15	–	–	≥20	–	–	19–16	–	–	≤15
**Pleuromutilin deriv.**										
Tiamulin (TIA)	30	≥9	–	–	–	–	–	≤8	–	–
**Sulphonamides**										
Trimethoprim – Sulfamethoxazole (SxT)	1.25/ 23.75	≥16	≥19	≥16	11–15	16–18	11–15	≤10	≤15	≤10

**Table 3 microorganisms-11-02410-t003:** The mean results of antimicrobial susceptibility testing for *A. pleuropneumoniae*, *S. suis*, and *P. multocida*, over 6-year period, classified as susceptible (S), intermediate (I), and resistant (R), expressed as percent (%) from the number of tested isolates.

The Antimicrobial	No. of Isolates Tested	*A. pleuropneumoniae*			No. of Isolates Tested	*S. suis*			No. of Isolates Tested	*P. multocida*		
		S % (No.)	I % (No.)	R % (No.)		S % (No.)	I % (No.)	R % (No.)		S % (No.)	I % (No.)	R % (No.)
Amoxicillin	133	95 (126)	2 (2)	4 (5)	207	95 (197)	1 (2)	4 (8)	119	97 (115)	2 (2)	2 (2)
Ampicillin	136	94 (128)	1 (1)	5 (7)	206	95 (196)	0 (1)	4 (9)	121	96 (116)	1 (1)	3 (4)
Amoxicillin + clavulanic acid	136	99 (135)	1 (1)	0 (0)	206	98 (201)	2 (4)	0 (1)	121	98 (119)	1 (1)	1 (1)
Penicillin	128	95 (122)	0 (0)	5 (6)	207	92 (191)	0 (1)	7 (15)	120	97 (116)	0 (0)	3 (4)
Tetracycline	123	76 (94)	6 (7)	18 (22)	205	20 (41)	5 (11)	75 (153)	106	85 (90)	1 (1)	14 (15)
Trimethoprim - Sulfamethoxazole	137	91 (125)	1 (2)	7 (12)	206	84 (174)	2 (4)	14 (28)	118	93 (110)	2 (2)	5 (6)
Ceftiofur	135	99 (134)	1 (1)	0 (0)	204	99 (202)	0 (1)	0 (1)	119	98 (117)	0 (0)	2 (2)
Oxytetracycline	137	76 (104)	4 (6)	20 (27)	203	22 (44)	6 (12)	72 (147)	109	84 (92)	3 (3)	13 (14)
Enrofloxacin	137	92 (126)	3 (4)	5 (7)	207	83 (171)	9 (19)	8 (17)	114	96 (109)	4 (5)	0 (0)
Marbofloxacin	132	92 (122)	3 (4)	5 (6)	206	91 (188)	3 (7)	5 (11)	119	99 (118)	0 (0)	1 (1)
Florfenicol	136	93 (128)	3 (4)	3 (4)	191	97 (186)	1 (2)	2 (3)	108	97 (105)	0 (0)	3 (3)
Cefquinome	107	100 (107)	0 (0)	0 (0)	180	93 (167)	4 (7)	3 (6)	105	92 (97)	4 (4)	4 (4)
Doxycycline	127	83 (106)	6 (7)	11 (14)	199	29 (58)	19 (38)	52 (103)	109	90 (98)	6 (6)	5 (5)
Erythromycin	3	67 (2)	0 (0)	33 (1)	196	42 (83)	7 (14)	51 (99)	4	75 (3)	25 (1)	0 (0)
Gentamycin	133	35 (47)	20 (26)	45 (60)	1	100 (1)	0 (0)	0 (0)	115	95 (83)	9 (8)	10 (11)
Tilmicosin	132	58 (76)	3 (4)	39 (52)	47	23 (11)	4 (2)	72 (34)	119	62 (74)	8 (10)	29 (35)
Streptomycin	131	18 (24)	5 (7)	77 (101)	54	0 (0)	2 (1)	98 (53)	113	35 (39)	3 (3)	63 (71)
Tiamulin	20	60 (12)	10 (2)	30 (6)	7	57 (4)	0 (0)	43 (3)	8	38 (3)	0 (0)	63 (5)
Apramycin	–	–	–	–	–	–	–	–	18	50 (9)	11 (2)	39 (7)
Neomicin	4	100 (4)	0 (0)	0 (0)	2	50 (1)	0 (0)	50 (1)	102	62 (63)	19 (19)	20 (20)

**Table 4 microorganisms-11-02410-t004:** Antimicrobial resistance of 137 *A. pleuropneumoniae* isolated over 2017–2022, expressed as percent (%) from the number of tested isolates.

The Antimicrobial	No. of Resistant Isolates	2017	2018	2019	2020	2021	2022
		% (No.)	% (No.)	% (No.)	% (No.)	% (No.)	% (No.)
Amoxicillin	5	0 (0)	0 (0)	0 (0)	27 (4)	3 (1)	0 (0)
Ampicillin	7	8 (1)	0 (0)	0 (0)	27 (4)	0 (0)	4 (2)
Amoxicillin + clavulanic acid	0	0 (0)	0 (0)	0 (0)	0 (0)	0 (0)	0 (0)
Penicillin	6	0 (0)	0 (0)	0 (0)	27 (4)	4 (1)	2 (1)
Tetracycline	22	0 (0)	0 (0)	20 (2)	40 (6)	27 (8)	15 (6)
Trimethoprim–Sulfamethoxazole	12	0 (0)	14 (2)	10 (1)	20 (3)	7 (2)	7 (4)
Ceftiofur	0	0 (0)	0 (0)	0 (0)	0 (0)	0 (0)	0 (0)
Oxytetracycline	27	1 (8)	1 (7)	1 (10)	7 (47)	8 (27)	9 (16)
Enrofloxacin	7	0 (0)	0 (0)	0 (0)	13 (2)	7 (2)	5 (3)
Marbofloxacin	6	0 (0)	0 (0)	0 (0)	13 (2)	3 (1)	6 (3)
Florfenicol	4	0 (0)	0 (0)	0 (0)	7 (1)	7 (2)	2 (1)
Cefquinome	0	0 (0)	0 (0)	0 (0)	0 (0)	0 (0)	0 (0)
Doxycycline	14	0 (0)	0 (0)	0 (0)	13 (2)	15 (4)	16 (8)
Erythromycin	1	0 (0)	0 (0)	0 (0)	0 (0)	0 (0)	100 (1)
Gentamycin	60	8 (1)	57 (8)	0 (0)	50 (7)	70 (21)	44 (23)
Tilmicosin	52	46 (6)	64 (9)	0 (0)	71 (10)	39 (11)	30 (16)
Streptomycin	101	62 (8)	79 (11)	0 (0)	93 (14)	100 (28)	78 (40)
Tiamulin	6	0 (0)	0 (0)	0 (0)	0 (0)	25 (1)	71 (5)
Neomicin	0	0 (0)	0 (0)	0 (0)	0 (0)	0 (0)	0 (0)

**Table 5 microorganisms-11-02410-t005:** Antibiotic resistance of *S. suis* over 2017–2022, expressed as percent (%) from the number of tested isolates.

The Antimicrobial	No. of Resistant Isolates	2017	2018	2019	2020	2021	2022
		% (No.)	% (No.)	% (No.)	% (No.)	% (No.)	% (No.)
Amoxicillin	8	5 (1)	0 (0)	7 (1)	0 (0)	4 (2)	6 (4)
Ampicillin	9	0 (0)	17 (4)	0 (0)	0 (0)	4 (2)	5 (3)
Amoxicillin + clavulanic acid	1	0 (0)	4 (1)	0 (0)	0 (0)	0 (0)	0 (0)
Penicillin	15	5 (1)	17 (4)	7 (1)	6 (2)	5 (3)	6 (4)
Tetracycline	153	95 (20)	79 (19)	71 (10)	71 (22)	80 (44)	63 (38)
Trimethoprim–Sulfamethoxazole	28	19 (4)	4 (1)	21 (3)	10 (3)	16 (9)	13 (8)
Ceftiofur	1	0 (0)	4 (1)	0 (0)	0 (0)	0 (0)	0 (0)
Oxytetracycline	147	86 (18)	74 (17)	64 (9)	71 (22)	80 (43)	63 (38)
Enrofloxacin	17	19 (4)	8 (2)	7 (1)	13 (4)	2 (1)	8 (5)
Marbofloxacin	11	10 (2)	4 (1)	8 (1)	13 (4)	2 (1)	3 (2)
Florfenicol	3	0 (0)	0 (0)	14 (2)	0 (0)	0 (0)	2 (1)
Cefquinome	6	0 (0)	4 (1)	7 (1)	0 (0)	2 (1)	7 (3)
Doxycycline	103	76 (16)	63 (15)	71 (10)	48 (15)	42 (22)	44 (25)
Erythromycin	99	58 (11)	54 (13)	54 (7)	42 (11)	46 (24)	53 (33)
Streptomycin	53	0 (0)	0 (0)	0 (0)	100 (1)	96 (24)	100 (28)
Tiamulin	3	0 (0)	0 (0)	0 (0)	0 (0)	33 (1)	50 (2)
Tilmicosin	34	0 (0)	0 (0)	0 (0)	0 (0)	81 (13)	68 (21)
Neomicin	1	0 (0)	0 (0)	0 (0)	0 (0)	50 (1)	0 (0)
Gentamycin	0	0 (0)	0 (0)	0 (0)	0 (0)	0 (0)	0 (0)

**Table 6 microorganisms-11-02410-t006:** Antibiotic resistance of *Pasteurella multocida* over 2017–2022, expressed as percent (%) from the number of tested isolates.

The Antimicrobial	No. of Resistant Isolates	2017	2018	2019	2020	2021	2022
		% (No.)	% (No.)	% (No.)	% (No.)	% (No.)	% (No.)
Amoxicillin	2	0 (0)	6 (1)	0 (0)	0 (0)	0 (0)	4 (1)
Ampicillin	4	0 (0)	6 (1)	13 (1)	0 (0)	0 (0)	7 (2)
Amoxicillin + clavulanic acid	1	0 (0)	0 (0)	0 (0)	0 (0)	0 (0)	3 (1)
Penicillin	4	11 (1)	6 (1)	0 (0)	0 (0)	3 (1)	3 (1)
Tetracycline	15	11 (1)	11 (2)	25 (2)	18 (2)	22 (7)	4 (1)
Trimethoprim–Sulfamethoxazole	6	11 (1)	0 (0)	29 (2)	10 (2)	3 (1)	0 (0)
Ceftiofur	2	0 (0)	0 (0)	0 (0)	5 (1)	0 (0)	3 (1)
Oxytetracycline	14	22 (2)	0 (0)	29 (2)	18 (2)	19 (7)	3 (1)
Enrofloxacin	0	0 (0)	0 (0)	0 (0)	0 (0)	0 (0)	0 (0)
Marbofloxacin	1	0 (0)	0 (0)	0 (0)	5 (1)	0 (0)	0 (0)
Florfenicol	3	0 (0)	11 (2)	13 (1)	0 (0)	0 (0)	0 (0)
Cefquinome	4	0 (0)	6 (1)	38 (3)	0 (0)	0 (0)	0 (0)
Doxycycline	5	0 (0)	0 (0)	14 (1)	6 (1)	9 (3)	0 (0)
Erythromycin	0	0 (0)	0 (0)	0 (0)	0 (0)	0 (0)	0 (0)
Gentamycin	11	29 (2)	0 (0)	25 (2)	6 (1)	8 (3)	11 (3)
Tilmicosin	35	38 (3)	22 (4)	50 (4)	25 (5)	36 (13)	21 (6)
Streptomycin	71	100 (7)	47 (8)	100 (8)	45 (9)	74 (26)	50 (13)
Neomicin	20	29 (2)	19 (3)	33 (2)	22 (2)	17 (6)	18 (5)
Apramycin	7	0 (0)	100 (1)	0 (0)	67 (2)	0 (0)	29 (4)
Tiamulin	5	0 (0)	0 (0)	0 (0)	0 (0)	50 (2)	100 (3)

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
