# Peer review of "Antimicrobial Resistance of *Actinobacillus pleuropneumoniae*, *Streptococcus suis*, and *Pasteurella multocida* Isolated from Romanian Swine Farms"

_microorganisms, 2023, doi:10.3390/microorganisms11102410_

Round 1
Reviewer 1 Report
Microorganisms – ID 2594502:
Antimicrobial resistance of Actinobacillus pleuropneumoniae, Streptococcus suis and Pasteurella multocida isolated from Romanian swine farms.
Authors evaluated 465 bacterial isolates including strain of Actinobacillus pleuropneumoniae, Streptococcus suis and Pasteurella multocida from swine farms from 2017 to 2020 and tested their antimicrobial resistance profiles. Interesting trends of AMR in the last 6 years among relevant antimicrobials, both for animal and human medicine, emerged.
Although the entire study is well written and structured and add valuable information about antimicrobial resistance trend of relevant bacteria for swine and public health, the entire section of discussion does not reflect the big effort done by authors in the evaluation of AMR profiles of bacteria strains.
Below, I included some suggestions.
Abstract:
Line 11: “465 samples”. Here authors define samples, whereas in the main text write about bacterial strains. Did from 465 samples authors isolate 465 strains? Please, revise accordingly.
Introduction:
Line 50: Streptococcus suis (S. suis) should be in italic. Please revise.
Table 1:
The acronym “APP” appears for the first time. I suggest authors to define and include it before, in the introduction. In alternative, a caption coulb be added below table 1.
Discussion:
This long section should be re-writed. It seems only a comparison with other studies in other country, along with the repetition of obtained results. Authors should discussed theri results, in the context of the most significant health and production problems of swine industry, stressing the on use of antimicrobials in veterinary and human medicine.
Authors could discuss also about the european survellaince on antimicrobial resistant bacteria and the restrinction on use of antimicrobials in veterinary medicine.
Author Response
Here I attached my reply to the review report for: Antimicrobial resistance of Actinobacillus pleuropneumoniae, Streptococcus suis and Pasteurella multocida isolated from Romanian swine farms
Thank you!
Madalina Siteavu

Reviewer 2 Report
The authors presented the changes in antimicrobial susceptibility evolution of A. pleuropneumoniae, S. suis and P. multocida isolated from pigs in Romania, from 2017 to 2022.
There is always interest in monitoring antimicrobial susceptibility of clinical isolates, so the work is of clinical significance.
Major issues.
Source of the isolates are not defined. This must be corrected as it provides extra information regarding the situation in the field. Please clarify a) location of farms, b) animals with signs or no, c) lungs with lesions or no, d) administration of antibiotics and types of antibiotics given, e) vaccination schedules in farms.
Please use EUCAST standards, not CLSI standards. Romania is a member of the EU, they pay for the EUCAST laboratories, they must use as well and this will allow comparison with relevant findings in other European countries. Serious mistake.
Minor issues.
Tables 3-6 can be transferred to supplementary material. The figures suffice in the results.
Discussion. 1) please separate in sub-sections, 2) please add comments in relation to data requested above to be added.
Overall. The manuscript must be corrected and re-evaluated. Final recommendation will be issued after re-reviewing.
Author Response

(The authors gave the same response as above.)

Round 2
Reviewer 1 Report
Authors complied with all reviewer suggestions and I appreciate the effort to improve the quality of the manuscript. I would not include any other comments or editing suggestions.
Author Response
We want to thank you for the time you put in reviewing our paper and the contributions you made with your suggestions.
Reviewer 2 Report
The authors have not responded well to all the points raised in the previous evaluation. The following issues still need to be addressed.
1. Please provide the exact locations of the farms within Romania. Mentioning just the country is not satisfactory and leads to suspicions regarding the study. Preferably a map will support the claims of the authors.
2. Use of American standards in Europe is very much against the core the European unification. Why the authors don't they also use feet and yards and miles to measure distances?
Please recalculate the findings by using European measures and please make all necessary changes in the manuscript.
This is not something difficult: you have the results, you have the standards, you can recalculate within 3 days.
Omission to recalculate with European standards will make the manuscript look fishy.........
